# Pragmatic cluster randomised cohort cross-over trial to determine the effectiveness of bridging from emergency to regular contraception: the Bridge-It study protocol

Sharon Tracey Cameron [1,2] Paula Baraitser,[3] Anna Glasier,[1] Lisa McDaid,[4] John Norrie,[5] Andrew Radley [6,7] Judith M Stephenson,[8] James Trussell,[9] Claire Battison,[5] Sarah Cameron,[10] Kathleen Cowle,[11] Mark Forrest,[10] Richard Gilson,[12] Beatriz Goulao,[10] Anne Johnstone,[1] Alison McDonald,[10] Alessandra Morelli [3] Susan Patterson,[4] Deirdre Sally,[12] Nicola Stewart,[12] The Bridge-It Study Group

For numbered affiliations see end of article.

**Correspondence to**
Professor Sharon Tracey Cameron;
sharon.cameron@ed.ac.uk

## ABSTRACT

**Introduction** Oral emergency contraception (EC) can prevent unintended pregnancy but it is important to start a regular method of contraception. Women in the UK usually access EC from a pharmacy but then need a subsequent appointment with a general practitioner or a sexual and reproductive health (SRH) service to access regular contraception. Unintended pregnancies can occur during this time.

**Methods and analysis** Bridge-It is a pragmatic cluster randomised cohort cross-over trial designed to determine whether pharmacist provision of a bridging supply of a progestogen-only pill (POP) plus rapid access to a local SRH clinic, results in increased uptake of effective contraception and prevents more unintended pregnancies than provision of EC alone. Bridge-It involves 31 pharmacies in three UK regions (London, Lothian and Tayside) aiming to recruit 626–737 women. Pharmacies will give EC (levonorgestrel) according to normal practice and recruit women to both intervention and the control phases of the study. In the intervention phase, pharmacists will provide the POP (desogestrel) and offer rapid access to an SRH clinic. In the control phase, pharmacists will advise women to attend a contraceptive provider for contraception (standard care). Women will be asked 4 months later about contraceptive use. Data linkage to abortion registries will provide abortion rates over 12 months. The sample size is calculated on the primary outcome of effective contraception use at 4 months (yes/no) with 90% power and a 5% level of significance. Abortion rates will be an exploratory secondary analysis. Process evaluation includes interviews with pharmacists, SRH clinicians and women. Cost-effectiveness analysis will use a healthcare system perspective and be expressed as incremental cost-effectiveness ratio.

**Ethics and dissemination** Ethical approval was received from South East Scotland REC June 2017. Results will be published in peer-reviewed journals and conference presentations.

## Strengths and limitations of this study

► Examines the important outcome of abortion rates over 1 year as an exploratory secondary analysis.
► Applicable only to women receiving levonorgestrel emergency contraception (EC), followed by a desogestrel progestogen-only pill (POP).
► Not applicable to use of ulipristal acetate for EC, since hormonal methods of contraception, such as the desogestrel POP, may interact with efficacy of ulipristal acetate, if started within 5 days.

**Trial registration number** ISRCTN70616901.

## INTRODUCTION

Unintended pregnancy is widely perceived as a major public health problem. Unintended pregnancy commonly ends in abortion and the UK has among the highest abortion rates in Europe.[1] In 2017, almost 200 000 pregnancies ended in induced abortion.[2 3] Unintended pregnancy also ends in childbirth; around 10% of UK births are unintended and 25% mistimed.[4] Unintended pregnancy is costly to the National Health Service (NHS) (estimated to cost over £1 billion annually)[5] and can be distressing for women. Unintended pregnancies are more common in young women from deprived backgrounds, contributing to widening health inequalities for both mother and baby, and their families.[2 3] Unintended childbirth can have both socioeconomic consequences for women and their families and mental health consequences.[6]

Oral emergency contraception (EC) prevents pregnancy in individual women following unprotected sex or contraceptive accidents. EC is only effective if taken before ovulation as it works by inhibiting or delaying ovulation.[7] Since EC became available from pharmacies in the UK without the need for a prescription, there has been a change in the pattern of access such that women who seek EC now choose to obtain this from a pharmacy rather than a contraceptive provider such as a general practitioner (GP) or sexual and reproductive health (SRH) service.[8] Although trials have shown that this facilitates access to EC and increases use, they have failed to show that this reduces unintended pregnancy rates within the population.[9]

There are two types of EC: the most widely used EC contains the progestogen levonorgestrel and should be taken within 72 hours of sex; the other EC contains the progesterone receptor modulator ulipristal acetate and should be taken within 120 hours of sex.[10] Neither formulation of EC prevents conceptions from subsequent acts of sex and the risk of pregnancy is increased up to threefold among women who have further unprotected sex in the same menstrual cycle after using EC than those who do not.[10] An effective method of contraception should, therefore, be started as soon as possible.[10 11] However, the only contraceptives that can be obtained from any pharmacy without a prescription are condoms, which have high failure rates.[12] This means that women usually need to make an appointment with a contraceptive provider (GP or SRH) and may experience delays in accessing regular contraception or lose the motivation to access a regular method altogether, which in turn may result in unintended pregnancies. In addition, although pharmacists in the UK are supposed to advise women on where to obtain ongoing contraception after EC, in one study fewer than half of pharmacists did so.[13] It is possible that if pharmacists could supply a temporary (bridging) method of contraception to women along with EC, this would bridge the gap until women could get an appointment with a contraceptive provider for contraceptive advice and supplies. The progestogen-only pill (POP) is an effective method of contraception with few contraindications[14] making it safer than the combined contraceptive pill/oral contraceptives for pharmacy provision. However, studies have shown that starting hormonal contraception containing a progestogen within 5 days of ulipristal acetate may reduce the efficacy of EC and so only EC containing levonorgestrel is suitable for use in conjunction with a bridging method of hormonal contraception in this way.[15 16]

### Pilot

In a pilot study in Edinburgh of 168 women presenting for EC,[17] 11 pharmacies were randomised to one of three groups to provide EC (levonorgestrel) and either (1) standard advice on where to obtain ongoing contraception or (2) 1 month of a POP or (3) the offer of rapid access to a local SRH service. Participants were

contacted by telephone 6–8 weeks later to determine their current contraceptive use. Compared with standard care, the proportion of women using effective contraception was significantly greater in both the POP (56% vs 16%, p=0.001) and the rapid access groups (52% vs 16%, p=0.027). This suggests that a supply of 1 month of POP after EC or rapid access from a pharmacy to an SRH service might increase short-term uptake of effective contraception following EC.

We now propose a large randomised trial to determine whether a pharmacy-based intervention designed to facilitate the uptake of effective contraception after EC increases use of effective contraceptive methods including the most effective long-acting reversible contraceptive methods (LARC) such as the contraceptive implant and intrauterine contraception[12] at 4 months when compared with standard care. We will examine contraceptive uptake at 4 months as most POP preparations are packaged as a 3-month supply and so by 4 months the pharmacy provided supply will have ended.

### Aim

The aim is to develop a simple and affordable intervention which facilitates the uptake of effective ongoing contraception among women obtaining EC from pharmacies thereby reducing unintended pregnancy. The primary objective is to determine whether offering women attending a pharmacy for EC, a 3-month bridging supply of POP plus the offer of rapid access to a local SRH service results in increased uptake of effective contraception. The study POP (desogestrel) is commonly used in the UK. In contrast to other POP s, the desogestrel POP reliably inhibits ovulation and has similar effectiveness to a combined hormonal contraceptive pill/oral contraceptives (COCP), yet fewer contraindications than a COCP.[14 15] This combined intervention (POP plus rapid access) offers both a highly safe temporary method of contraception and facilitates access to a specialist contraceptive service where all methods of contraception including the most effective LARC methods can be provided. If this intervention leads to increased uptake of effective contraception including LARC methods compared with standard care alone then we might expect that this would translate into fewer unintended pregnancies for women.

### METHODS AND ANALYSIS
### Study design and setting

A pragmatic cluster randomised cohort cross-over trial with cost-effectiveness including process, outcome and economic evaluation involving 31 pharmacies in three UK regions (15 in London (South and Central), 12 in Lothian (Edinburgh and region) and 4 in Tayside (Dundee and region).

### Patient and public involvement

The members of the patient and public involvement (PPI) group at the SRH service in Edinburgh contributed

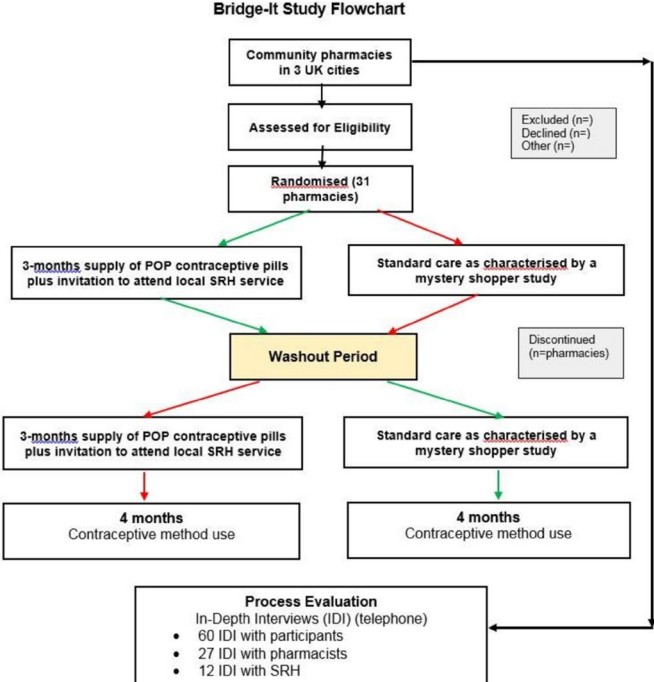

**Figure 1** Bridge-it flow chart. POP, progestogen-only pill; SRH, sexual and reproductive health.

to the design of this study. The study protocol and documentation were reviewed and approved by the chair and members of the PPI group. The plain English summary was edited by a PPI member and improved as a result. There are three PPI members that participate and contribute to the Bridge-It Trial steering Committee meeting that provides oversight of the study. PPI group will be involved in the dissemination of the study results.

### Intervention
The planned intervention is a composite intervention. Each woman in the intervention phase will receive 3 months of POP (in a single package covering 3 months) and an invitation to attend a local participating SRH service to discuss and obtain effective contraception (including LARC methods). Three packets of POP, (75 mcg desogestrel; UK) containing 28 tablets each will be provided at no cost to women as a bridging method of contraception, providing them with 3 months of contraception during which time they can get an appointment with a contraceptive provider to obtain their preferred method of contraception. Locally approved patient group directions (ie, strict criteria to permit provision of specified medicines by non-prescribers) will permit participating pharmacists to dispense the POP to women recruited to the study. Prestudy training will be undertaken with participating pharmacists including identifying medical contraindications to POP, potential drug interactions medications and 'missed pill' guidance for POP. Pharmacists will advise women to start the POP the day following intake of EC[10] and provide women with a patient information booklet on the POP from the family planning association (www.fpa.org).

Pharmacists will encourage women to attend the participating local SRH service to obtain the contraceptive method of their choice. Participants (intervention phase) will be given a study card to alert staff at SRH services that they are in the Bridge-It study and should be seen as a drop in for contraception that same day. This card will also provide written information about the location and opening hours of the local participating SRH service. These SRH services are within a 5 mile radius of the participating pharmacies and provide all methods of contraception at no cost as is the norm in the NHS.

### Standard care
A mystery shopper exercise[13] will be undertaken in all 31 participating pharmacies to characterise 'standard care' (usually verbal advice to visit a clinic for contraception, with/without written and verbal information) in the control phase. The mystery shopper visits will be conducted when the pharmacy is not recruiting and just before the control phase starts. The mystery shoppers and the scenario used will be chosen by the PPI group. A simple scenario relating to request for EC will be used. Immediately after leaving the pharmacy, the mystery shopper will complete a standard data collection proforma, recording any information given by the pharmacist about use of contraception after taking EC, including provision of written information.

### Participants
We will recruit a total of 626–737 women presenting for EC. The final number will be determined based on the observed ratio of the between-period within-cluster correlation (BPC) and the within-period within-cluster correlation (WPC)—with the larger sample size (near the 737 upper limit) required for values of BPC close to zero, and the smaller sample size (near the 626 lower limits) required for values of BPC close to WPC.[18] Each pharmacy will be expected to recruit an average of 12–13 women to the intervention arm and 12–13 women to standard care. This allows for a 25% lost to follow-up at 4 months (missing data on primary outcome).

### Randomisation
Each pharmacy will be randomised to either the intervention phase for approximately 20 weeks followed by standard care phase for 20 weeks or vice versa with a washout period of 2 weeks between the two phases. The order in which pharmacy allocation to each arm is undertaken (intervention or control first) will be randomised (figure 1). The order of delivery of intervention or control for each pharmacy was randomised for this cluster crossover design from a randomisation file prepared by the study statistician in the Data Centre at the Centre for Healthcare Randomised Trials (CHaRT), University of Aberdeen, using SAS V.6.4 for Windows. The method used for generating the random unpredictable mix of permuted blocks was a computer software algorithm that

## Box 1  Inclusion and exclusion criteria

**Inclusion criteria**
► Intake of EC (levonorgestrel).
► Capacity to give informed consent to participate in the trial which includes adherence to trial requirements.
► Age 16 years or over.
► Willing to give contact details and be contacted at 4 months by phone or text or email or post.
► Willing to give identifying data sufficient to allow data linkage with NHS registries.

**Exclusion criteria**
► Contraindications to the progestogen-only pill (POP).
► On medication that interacts adversely with POP.
► Already using a hormonal method of contraception.
► Require interpreting services.
► If pharmacist has concerns about non-consensual sex.
EC, emergency contraception; NHS, National Health Service.

randomly allocated blocks of size 2, 4 and 6; blocking was used to ensure balanced group sizes.

This is a cluster cohort cross-over design so it is the pharmacy that is the unit of randomisation and the 'cross-over' means that we are just randomising the order that each pharmacy gives the intervention in. The 'cohort' label means that we expect different women to be recruited within each site in the two periods (intervention and control phases).

### Recruitment

The pharmacist will assess medical eligibility of women presenting for EC for the study, provide EC according to normal practice and invite eligible women to participate. A detailed patient information sheet will be provided to all women and informed consent will be obtained by participating community pharmacists. The EC used in this study is levonorgestrel and will be given in the clinically indicated dose for the woman's weight (1.5 mg or 3 mg levonorgestrel).[10]

Inclusion and exclusion criteria are shown in box 1. Women who give written consent will be recruited in the study. We recognise the importance of participant retention and will offer a voucher of £10 at recruitment.[19]

### Outcomes

A full list of study outcome measures is included in table 1. Outcomes at 4 months will primarily be collected via telephone interviews or via web-based questionnaires. However, participants will also have the option to provide the same information by a postal questionnaire. The primary outcome is use of effective contraception at 4 months (intervention vs standard care).

Secondary outcomes are proportion of participants having an abortion within 12 months of EC use using record linkage from participants to national registries and cost-effectiveness.

**Table 1**  Study outcomes

|  | Data source |
|---|---|
| **Main outcome** |  |
| Use of effective contraceptive method (hormonal or intrauterine) in intervention versus control at 4 months | Self-reported (telephone or self completed survey) at 4 months |
| **Secondary outcomes** |  |
| Numbers undergoing an abortion within 12 months intervention versus control | National abortion registries |
| Economic evaluation | Incremental cost-effectiveness ratio for pregnancies prevented |

### Process evaluation measures

A process evaluation will be conducted as part of the study to assess potential issues concerning intervention implementation, the causal mechanisms of impact and the contextual factors that could affect these. The process evaluation will comprise of quantitative and qualitative data measures, as detailed in box 2.

### Data collection

#### Quantitative data

Participant flow: Participant flow through the study will be assessed and reported following the Consolidated Standards of Reporting Trials flow chart.

#### Baseline

Participant demographics and reproductive history is collected at recruitment by a self-administered paper questionnaire given to them by the pharmacist. Demographic data will also be reported for the process evaluation, protocol adherence checklists and for recruitment screening forms (see box 2).

#### Contraceptive use at 4 months

This will be based on self-reported data from women at a telephone follow-up interview with a research nurse at 4 months after obtaining EC. If participants prefer, the questions can be self-completed by a web-based questionnaire or paper questionnaire sent by the post. Women will be asked what method of contraception they are using (if any), if they attended a GP or SRH service for this, if they used the POP (intervention phase only) and their pregnancy status. If pregnancy has occurred since EC then the validated London Measure of Unintended Pregnancy tool will be administered to measure intended-ness of the index pregnancy[20] (table 1).

#### Abortion rates at 12 months

Information Services Division (Scotland) and Department of Health (England) will provide the number of abortions occurring during the 12-month follow-up

**Theory**
Theory of change model

**Study team**
Pharmacy recruitment forms (study team members involved in recruitment will routinely record decision-making contributing to pharmacy selection, including: number of contacts made; responses from potential pharmacists; rationales for inclusion/exclusion and reasons for refusal).

**Pharmacists**
Participant observation of training and review of training and intervention materials.
Recruitment monitoring forms (n=100% of pharmacists) and protocol adherence checklists (n=100% of pharmacists).
Follow-up semistructured telephone interviews with pharmacists (n=27; one with each pharmacy involved).

**Sexual and reproductive health (SRH) providers**
Semistructured telephone interviews with SRH providers (n=12; with service manager, mix staff at 2x services in London; 1x service in Edinburgh; 1x service in Dundee).

**Participants**
Telephone questionnaire administered by research nurse at 4 months postintervention (n=100% participants).
Semistructured telephone interviews at 4 months postintervention (n=60; 22 in London, 30 in Edinburgh and 8eight in Dundee).

**Context**
Audit of local contraceptive services within 10 miles of study sites in London, Edinburgh and Dundee.
Monitoring of contemporaneous events, such as relevant high coverage media stories using Google Alerts.

period in each arm by conducting linkage of the identifiers (collected at baseline) from study participants.

### Validation

We will check with data from local SRH services to determine the numbers of participants from intervention and control phases who attend the local participating SRH service, and which method of contraception they received.

### Qualitative data

Semistructured, qualitative telephone interviews of a purposive sample of up to 60 women who received the intervention (approximately 22 in London, 30 in Edinburgh and 8 in Dundee) will be undertaken. Participants who consent to these telephone interviews will be contacted by the Process Evaluation Research Assistant. Interviews will explore experience of intervention acceptability in more depth and assess experiences of bridging from EC to regular contraception, and reasons for doing so or not (Box 2). Interviews will be conducted soon after the 4-month follow-up.

Interviews with 27 pharmacists and 12 SRH service providers will explore their perceptions of barriers and facilitators to implementation and more broadly, their views on the intervention, the trial and the target

population. Interviews will be conducted by the process evaluation research assistant soon after the intervention phase has completed.

For the process evaluation, data collection also includes: review of training and materials, observation of training, mapping of local contraceptive services within 10 miles of study sites and monitoring of contemporaneous events, such as relevant high coverage media stories using Google Alerts (box 2).

### Sample size calculation

The study is a pragmatic cluster randomised cohort cross-over trial. Ideally the control and intervention phases should be of roughly equal duration and size, and the sample size is calculated assuming an equal cluster size in both control and intervention periods, and equal across sites (the expected site/period average). In practice, there is variability in EC demand across sites and over time, for example, demand is affected by peak holiday periods (recruitment decreases) and student term times (recruitment increases); and the ability of a pharmacy to translate that variable demand into study recruits depends on many factors, including changes to circumstances at individual pharmacies or loss of/change of pharmacists at multipharmacist stores.

Informed by our pilot study, we have assumed that effective contraception use in the control would be 30% and we were likely to achieve a 50% relative improvement to 45%. This means the sample size is in the range 626–737, assuming 25% of women do not provide 4-month contraceptive use data, and an average cluster size of 12–13 in each period, and around 25 pharmacies taking part, with 90% power and a 5% level of significance. The uncertainty in the required sample size rests on the assumed BPC and its relationship to the other component of variability, the WPC.[18] The WPC is the usual correlation (known as the intraclass correlation in a standard parallel groups non-cross-over cluster setting) of two individuals' outcomes within a cluster (in the same period). The BPC on the other hand is the correlation between two individuals' outcomes in the same cluster between the two periods. If the BPC is zero, there is no advantage in a cross-over design over the standard parallel groups cluster design; if the BPC equals the WPC then the crossover is as efficient as an individually randomised design. We will finalise the sample size depending on the observed ratio of the BPC to WPC, and the observed rate of attrition, but still assuming the same control rate and treatment effect, once we have 4-month data on at least 500 participants.

### Quantitative analyses

There will be a single analysis at study end (there is no opportunity for any interim analyses given the cross-over design) although an independent data monitoring committee will monitor study progress and any safety issues. This will follow the intention-to-treat principle and will use a hierarchical model appropriate for the specific outcome. For the primary outcome, this will be a mixed-effects

 

logistic regression, using the hierarchical model approach as recommended by Turner *et al* for a cluster cross-over design.[21] We will prespecify any individual level (or cluster level) covariates that we intend to adjust for, and the comprehensive statistical analysis plan will specify the sensitivity type analyses that will explore how robust the findings are to any missing data at the cluster level (probably unlikely) and the individual level (expected to be substantial for the patient reported outcomes at 4 months). As well as the usual assumption of missing at random, we will try to explore possible mechanisms for non-ignorable (informative missingness) at the individual level, which may well be operating in this context. Subgroup analyses will explore the possible effect modification by LARC (most effective contraceptive methods) versus non-LARC vs no use of contraception.

### Qualitative and mixed-methods process evaluation analyses

All process data will be analysed independently of the outcome data and, importantly, documented before the outcomes are known. Qualitative analysis of in-depth interviews will be recorded and transcribed verbatim. Transcription and analysis (proceeding case by case) will start with the first interview and be ongoing during the course of data collection, allowing for emergent themes to be identified and explored in future interviews. The transcripts will be read repeatedly and coded for analysis. Data management will be assisted by the software, QSR NVivo V.10. Analysis will be undertaken using 'Framework Analysis' a method of proven validity and reliability where data are coded, indexed and charted systematically, then organised using a matrix or framework.[22] Constant comparison will be carried out to ensure that the analysis represents all perspectives and negative ('deviant') cases.

The multisource process evaluation will be synthesised to address the three key process evaluation questions: (1) what was delivered, (2) how it was delivered and (3) what role context may have had in shaping the delivery/outcomes

### Economic evaluation

An economic evaluation will be undertaken comparing the intervention and control arms in a cost-effectiveness analysis. A trial-based analysis will be followed by the construction of a decision model to extrapolate future costs and benefits beyond the completion of the trial. The overall perspective used will be that of the health system. Costs will include the pharmacist training to provide POP, direct and indirect costs of health service use, and the provision and dispensing of POP. We will compare the costs to the NHS in the intervention and control arms. To account for differences in the numbers of women in the two arms, we will compare the cost per woman in each arm. In the control arm, the costs are (1) cost of EC, (2) cost of pharmacist provision of EC, (3) cost of abortions. In the intervention arm, the costs are in addition to these (4) the cost of the POP, (5) cost of pharmacist training to provide POP and (6) cost of pharmacist provision of POP. The costs (1) and (2) are the same in both groups and so the extra cost of

the intervention will be the sum of (4), (5) and (6). The cost per women who has an abortion is the same in both groups except that we hypothesise that the abortion rate will be lower in the intervention group. We can then state the outcome as conventional incremental cost-effectiveness ratio that is, for every £100 spent on the intervention results in x fewer abortions for a savings of £Y. If Y is greater than 100 then the intervention is cost-effective. We will examine the sensitivity of the outcomes to variations in the costs of 4, 5 and 6.

### Data management and clinical trial unit support

Data will be collected on a paper case report form and will be entered directly into the trial database. Data will be entered into a trial database by pharmacists, research nurses or staff at the trial coordinating centre. The data management and statistical support (including responsibility of data and final dataset) for the study is provided by the UKCRC registered Clinical Trial Unit (CTU) the CHaRT at the University of Aberdeen while the trial management is provided by another UKCRC registered CTU, the Edinburgh Clinical Trials Unit (ECTU) at Edinburgh University.

### Ethics and dissemination

The Bridge-It trial involves procedures and medications, which are well established in current NHS clinical practice and use. Adverse events may occur during or after the use of EC or POP and are well documented in the POP patient information leaflet. Serious adverse events will be recorded at the 4-month follow-up interview and reported to the study sponsor. The study will be conducted in accordance with the principles of Good Clinical Practice.

Annual progress reports and a final report at the conclusion of the trial will be submitted to REC within the timelines defined in the regulations. Protocol modifications are communicated by the ECTU to study sites via email and electronic newsletters.

The Bridge-It study website will include trial materials, trial progress and summaries of key findings. In addition, public engagement and dissemination will also be undertaken via our PPI group.

The results of the study will be published in the academic journals and all participants will be offered a lay summary of the main findings of the study. The findings will also be presented at national and international conferences and disseminated via social media.

### Trial status

As at 7 February 2019, the study had recruited 503 participants across 29 sites. Recruitment to the first period completed on 13 January 2019, with 391 participants recruited at 29 sites. Recruitment is scheduled to be completed by June 2019, with analyses of the 4-month primary outcome expected to be available by October 2019.

### DISCUSSION

Unintended pregnancy is widely perceived as a major public health problem. The proposed intervention in the

Bridge-It study provides both temporary contraception (the POP) and facilitates access to effective contraception at a local SRH clinic. The cluster design was felt necessary for logistical reasons and confirmed in the qualitative work of our pilot study[17 23] that an individually randomised trial would simply not recruit, as it was not feasible for pharmacists within a busy pharmacy to take additional time to randomise each individual. The cross-over nature of the cluster design was chosen for efficiency, and by having a different set of women recruited at a pharmacy in the two different periods we avoided contamination by the participant. The purpose of the washoutout period was to minimise intervention effect carrying over from one period to another, as part of any contamination effect mediated by the pharmacist. With the cluster cross-over design, each cluster will act as its own control and fewer pharmacies are required than with a parallel cluster design.

If our proposed intervention works, then this could prevent more unintended pregnancies for more women. If the intervention is cost-effective then it could have cost savings for the NHS.

## Author affiliations
[1] Obstetrics and Gynaecology, University of Edinburgh, Edinburgh, UK
[2] Sexual and Reproductive Health, NHS Lothian, Edinburgh, UK
[3] Department of Sexual Health, King's College Hospital NHS Foundation Trust, London, UK
[4] MRC/CSO Social and Public Health Sciences Unit, University of Glasgow, Glasgow, UK
[5] Edinburgh Clinical Trials Unit, Usher Institute, University of Edinburgh, Edinburgh, UK
[6] Directorate of Public Health, NHS Tayside, Dundee, UK
[7] Division of Cardiovascular Medicines and Diabetes, Ninewells Hospital and Medical School, Dundee, UK
[8] UCL Elizabeth Garrett Anderson Institute for Women's Health, University College London, London, UK
[9] Office of Population Research, Princeton University, Princeton, New Jersey, USA
[10] Health Services Research Unit, University of Aberdeen, Aberdeen, UK
[11] Boots UK Ltd, Edinburgh, UK
[12] Institute for Global Health, University College London, London, UK

**Acknowledgements** Thank you to the members of the patient and public involvement group at the SRH service in Edinburgh who contributed to the design of this study. Thanks to the women who have participated in this study and to the community pharmacists, research staff and healthcare professionals at the local SRH services who have assisted with the implementation of this study. Thanks to Laura Flett for trial management support.

**Collaborators** The Bridge-It Study Steering Committee provides oversight for this trial on behalf of the sponsor (University of Edinburgh and NHS Lothian) and the funder. The members are: Professor Peter Brocklehurst (Chair), Birmingham Clinical Trials Unit; Dr Lucy Michie, Sandyford Glasgow; Professor Kaye Wellings, London School of Hygiene and Tropical Medicine; Joanna Loudon, PPI member, Edinburgh; Kirsten Stuart PPI member Edinburgh; Emily Whittaker PPI member Edinburgh. The Data Monitoring Committee is an independent multidisciplinary group consisting of clinicians and statisticians. The members are: Professor Claire Anderson (Chair), University of Nottingham; Professor Elizabeth Allen, London School of Hygiene and Tropical Medicine; Professor Caroline Moreau, Johns Hopkins Bloomberg School of Public Health, USA. A copy of the DMC charter is held in Edinburgh Clinical Trials Unit. The study has co-sponsorship between The University Court of the University of Edinburgh and Lothian Health Board. The sponsors representative is accord@nhslothian.scot.nhs.uk. The paper presents independent research funded by the National Institute for Health Research (NIHR).

**Contributors** STC, AG, JN, LM, AR, PB, JMS and JT developed the original protocol. STC, CB, MF, RG, AMD, BG, AJ, AM, SP, DS,NS and KC contributed to later stages of study design. STC wrote the draft of the protocol with significant input from all authors at all stages. All authors contributed, read and approved the final manuscript.

**Funding** The Bridge-It study is funded by the National Institute for Health Research's Health Technology Assessment Programme. HTA Project:15/113/01.

**Disclaimer** The views expressed are those of the author(s) and not necessarily those of the NIHR, the NHS or the Department of Health.

**Competing interests** AG is a member of HRA Pharma scientific advisory board. PB is a Clinical Director of the not-for profit community interest company SH:24 that provides online sexual health services in partnership with the NHS. AR received research grants, educational grants and consultancy with Gilead Research grants from Roche and BMS Educational grants from Abbvie. JN is Deputy Chair of the NIHR/HTA General Board Committee. NIHR/HTA funded this research.

**Patient consent for publication** Not required.

**Ethics approval** A favourable ethical opinion has been obtained from the South East Scotland REC in June 2017. Approvals have been obtained from NHS Research Scotland (NRS) and Health Research Authority (HRA) England prior to commencement of the study.

**Provenance and peer review** Not commissioned; externally peer reviewed.

**ORCID iDs**
Sharon Tracey Cameron http://orcid.org/0000-0002-1168-2276
Andrew Radley http://orcid.org/0000-0003-4772-2388
Alessandra Morelli http://orcid.org/0000-0002-9803-2136

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
