## [Reviewer comments · BMJ Open]

ARTICLE DETAILS

TITLE (PROVISIONAL)	A pragmatic cluster randomised cohort crossover trial to determine the effectiveness of bridging from emergency to regular contraception: The Bridge-It study protocol
AUTHORS	Cameron, Sharon; Baraitser, Paula; Glasier, Anna; McDaid, Lisa; Norrie, John; Radley, Andrew; Stephenson, Judith; Trussell, James; Battison, Claire; Cameron, Sarah; Cowle, Kathleen; Forrest, Mark; Gilson, Richard; Goulao, Beatriz; Johnstone, Anne; McDonald, Alison; Morelli, Alessandra; Patterson, Susan; Sally, Deirdre; Stewart, Nicola

VERSION 1 – REVIEW

REVIEWER	Rebecca Simmons University of Utah, United States of America
REVIEW RETURNED	26-Apr-2019

GENERAL COMMENTS	Task-shifting in sexual and reproductive health is an important support mechanism that has been shown to improve contraceptive access. Some states in the US and some countries have made short-acting, reversible methods (pills, patches, rings) available over-the counter to good effect. This study aligns with the ongoing research around increasing contraceptive access, through developing an intervention aimed at bridging the gap between emergency contraception and contraception. The majority of my comments are very minor - overall, this protocol is clear and I am hopeful the outcomes of the study will be used to convince public health and other health administrators in the UK to increase contraceptive access in pharmacies (or at least adapt some sort of protocol that increases linkage between clinics and pharmacies). However, a few things could be improved within the specific manuscript: 1) The framing of this protocol paper in the introduction seems to indicate that unintended pregnancy is a public health problem because unintended pregnancies lead to abortions. Abortion is not a disease outcome and, indeed, may be the most positive outcome possible for some women who experience an unintended pregnancy. Induced abortion conducted in facilities by trained providers is one of the safest medical obstetric procedures available. Studies like the Turn Away Study have demonstrated that women who receive abortions do not have an increased risk of mental health problems. Reducing unintended pregnancy through increasing service delivery access is a goal that aligns with both the human-rights framework of sexual and reproductive health and with tenants of reproductive justice: I would suggest the authors frame their intervention as such in this paper, rather than the focus on
---

	abortion prevention. 2) In a similar vein, it seems to me that the secondary analysis would identify ALL pregnancy outcomes, rather than simply abortion outcomes, even if the investigators are only relying on self-report from follow-up surveys. All outcomes are of interest and importance for measurement, particularly as the economic consequences of carrying an unwanted pregnancy to term have more costs associated with them. 3) I would suggest the authors further explain the rationale for pharmacists only dispensing progestin-pills, rather than the full range of short-acting methods (combined/progestin-only/patch/vaginal ring). Currently, the rationale is that progestin-only pills have fewer contraindications than COCs and thus, would be easier for pharmacists to dispense; however, there is no evidence that I am aware of supporting the idea that pharmacists are less capable of identifying contraindications for dispensing COCs than they are progestin-only pills. There IS evidence that the typical use efficacy of POC's is slightly lower, as this method has a higher use burden (as they need to be taken at the exact same time every day to maintain effect). Thus, as a bridge method, they may have limitations. I realize the trial is already underway and that this is what is being done; however, I would encourage the authors to think through a more thorough response to this limitation, as it will surely be identified by readers. 4) It would be good to spend a little time discussing the training given to pharmacists. It would be good to know whether or not the training included other aspects of contraceptive provision (aside from contraindications), such as effective methods of contraceptive counseling (since that is in essence what they are doing in limited form). 5) Did the pharmacists provide women with any materials around contraceptive methods during the intervention or control periods? Would be good to know whether women were given decision-aids or anything similar when they left the pharmacy. Overall, a good study. Good luck!
--	---

REVIEWER	Christiane Borges do Nascimento Chofakian University of São Paulo, Brazil
REVIEW RETURNED	14-May-2019

GENERAL COMMENTS	I assume that the manuscript is not an Original Research Article. It is a description of a study being carried out at the present time. Results just show some descriptions but no analysis that can be considered an evaluation.
---

REVIEWER	Tina Raine-Bennett Division of Research Kaiser Permanente Northern California
REVIEW RETURNED	20-Jun-2019

GENERAL COMMENTS	The purpose of the study is to determine whether pharmacist provision of a bridging supply of a progestogen only pill (POP) plus rapid access to a local SRH clinic, results in increased uptake of effective contraception and prevents more unintended pregnancies
--

	than provision of EC alone. This is a well-planned study which holds promise for answering important questions about optimizing care for women who seek emergency contraception at pharmacies. Comments on the manuscript:  1) The status of the study is provided at the end; i.e. the date enrollment started and anticipated study end; would be helpful to have this earlier in the manuscript (i.e. abstract) 2) It would be helpful if the introduction, the methods, or discussion included a section which describes consideration given to why cluster randomization is necessary and the risk of the intervention effect from one period carrying over to the control period. 3) An explanation of how the methods for obtaining the WPC and the BPC were chosen for the sample size calculation would be helpful for readers (i.e. why use 4-month data of 500 subjects?), and potential factors which might influence the magnitude of the variation for the estimates .
--	---

REVIEWER	Eric Vittinghoff University of California, San Francisco
REVIEW RETURNED	08-Jul-2019

GENERAL COMMENTS	Statistical review of Cameron et al, A cluster randomized controlled trial to determine the effectiveness of bridging from emergency to regular contraception: the Bridge-it study protocol. The authors have done an admirable job of designing and clearly explaining the statistical aspects of the proposed Bridge-it trial. I do have some minor concerns, questions and suggestions:  1. WPC and BPC are described as components of variability in the paragraph labeled participants, and then, more accurately (in my view), as correlations in the paragraph labeled sample size calculation. I suggest making these descriptions consistent. 2. It might be worthwhile to specify the range of values of these two correlations being considered. For what it's worth, my estimates of the power of the study for the proposed range of sample sizes were ~90% for values of WPC <5%, and BPC no more than 2 points less than WPC. 3. The updated sample size calculation will correctly use the hypothesized treatment effect, not an internal estimate. While it makes good sense using study data to update preliminary estimates of WPC and BPC after 4-month outcomes for 500 participants have been obtained, omitting treatment from the model used to estimate the variance components could inflate both correlations at least slightly, although this bias might be negligible compared to sampling error. Will the WPC and BPC estimates be obtained without unblinding, to avoid inflating the type-I error rate (or at any rate giving the appearance of having done so)? 4. The Bayesian approach of Turner et al. is developed in the context of multivariate outcomes and thus would apply to the secondary process outcomes that are to be collected. However, it is less clearly suited to the univariate primary outcome of contraceptive use at 4 months. One concern is that Bayesian methods may not ultimately be acceptable to reviewers expecting frequentist methods for an RCT, in part because, as Turner et al. point out, the results can be sensitive to the choice of priors. If Bayesian analysis of the primary outcome is intended, I think this should be made explicit. 5. In addition, the analysis plans should probably mention some standard checks for crossover trials such as checks for order effects and order-treatment interaction (recognizing that power for the latter is low).
--

	6. I didn't understand the proposal at the end of the paragraph on quantitative analyses for a subgroup analysis of effect modification by LARC vs non-LARC, which seem like two possible forms of the binary outcome of contraceptive use at 4 months, not baseline characteristics (as usual for effect modifiers). If so, wouldn't it make sense to analyze this as a multinomial outcome, comparing the treatment effects on LARC vs no use and non-LARC vs no use? 7. The plans for sensitivity analyses examining robustness to missing data, in particular non-ignorable missingness, are excellent, and will be important if the expected 25% of participants do not provide the primary outcome at 4 months. Still, that is a rather large loss to follow-up rate. Could something be done to reduce it, such as provision of stipends at 4 months as well as baseline?
--	--

VERSION 1 – AUTHOR RESPONSE

Reviewer(s)' Comments to Author:

Reviewer: 1

The majority of my comments are very minor - overall, this protocol is clear and I am hopeful the outcomes of the study will be used to convince public health and other health administrators in the UK to increase contraceptive access in pharmacies (or at least adapt some sort of protocol that increases linkage between clinics and pharmacies). However, a few things could be improved within the specific manuscript:

1) The framing of this protocol paper in the introduction seems to indicate that unintended pregnancy is a public health problem because unintended pregnancies lead to abortions. Abortion is not a disease outcome and, indeed, may be the most positive outcome possible for some women who experience an unintended pregnancy. Induced abortion conducted in facilities by trained providers is one of the safest medical obstetric procedures available. Studies like the Turn Away Study have demonstrated that women who receive abortions do not have an increased risk of mental health problems. Reducing unintended pregnancy through increasing service delivery access is a goal that aligns with both the human-rights framework of sexual and reproductive health and with tenants of reproductive justice: I would suggest the authors frame their intervention as such in this paper, rather than the focus on abortion prevention.

Response: we agree with the sentiments of the reviewer. We have reworded the relevant sentences to minimise the risk of misinterpreting.

'Unintended pregnancy is widely perceived as a major public health problem. Unintended pregnancy commonly ends in abortion and the UK has among the highest abortion rates in Europe'

2) In a similar vein, it seems to me that the secondary analysis would identify ALL pregnancy outcomes, rather than simply abortion outcomes, even if the investigators are only relying on self-report from follow-up surveys. All outcomes are of interest and importance for measurement, particularly as the economic consequences of carrying an unwanted pregnancy to term have more costs associated with them.

Response: The 4 -month follow up will determine self report of pregnancies that have ended by that time (abortion or miscarriage or ectopic) or are continuing at 4 months- it will be too early for births at 4 months.

Although we agree with the reviewer that all outcomes are of interest, for birth data at 12 months we would need permission to access maternity databases. The numbers of births in the cohort are expected to be small and the additional work/ permissions / analysis would not be practical nor justifiable within given resources.

3) I would suggest the authors further explain the rationale for pharmacists only dispensing progestin-pills, rather than the full range of short-acting methods (combined/progestin-only/patch/vaginal ring). Currently, the rationale is that progestin-only pills have fewer contraindications than COCs and thus, would be easier for pharmacists to dispense; however, there is no evidence that I am aware of supporting the idea that pharmacists are less capable of identifying contraindications for dispensing COCs than they are progestin-only pills. There IS evidence that the typical use efficacy of POC's is slightly lower, as this method has a higher use burden (as they need to be taken at the exact same time every day to maintain effect). Thus, as a bridge method, they may have limitations. I realize the trial is already underway and that this is what is being done; however, I would encourage the authors to think through a more thorough response to this limitation, as it will surely be identified by readers.
Response:

We wished a simple intervention. Whilst we do not deny that the COCP could be provided by pharmacists, nevertheless the COCP does have more risks associated with use and would require more pharmacy training / longer counselling of clients – on top of the time taken for the EC consultation. The POP has very few risks and so is easier to provide from a pharmacy setting. Furthermore, the POP we used (desogestrel) is a POP that reliably inhibits ovulation and same efficacy as combined pill.

We have added to the aim of the study to clarify this use of POP for the reader.

'The study POP (desogestrel) is commonly used in the UK. In contrast to other POP s, the desogestrel POP reliably inhibits ovulation and has similar effectiveness to a combined hormonal oral contraceptive pill (COCP), yet fewer contraindications than a COCP (14,15). This combined intervention (POP plus rapid access) offers both a highly safe temporary method of contraception and facilitates access to a specialist contraceptive service'

4) It would be good to spend a little time discussing the training given to pharmacists. It would be good to know whether or not the training included other aspects of contraceptive provision (aside from contraindications), such as effective methods of contraceptive counseling (since that is in essence what they are doing in limited form).

Response: Training was only provided on the POP and the study requirements (see intervention section). It is supposed to be 'standard care' that pharmacists who provide EC also discuss contraception and were to obtain it. We have clarified this in the introduction:

' . In addition, although pharmacists in the UK are supposed to advise women on where to obtain ongoing contraception after EC , in one study fewer than half of pharmacists did so [13]'

5) Did the pharmacists provide women with any materials around contraceptive methods during the intervention or control periods? Would be good to know whether women were given decision-aids or anything similar when they left the pharmacy.

Response: The mystery shopper visit was to 'describe this standard care' and what information was given. This will be reported in the final study results.

In the intervention phase a FPA booklet on the POP was provided with supplies of POP. We have added this to the intervention section:

'..and provide women with a patient information booklet on the POP from the family planning association (www.fpa.org).

Reviewer: 2

1.1 assume that the manuscript is not an Original Research Article. It is a description of a study being carried out at the present time. Results just show some descriptions but no analysis that can be considered an evaluation.

Response: Correct- this is a protocol for a study

Reviewer: 3

The purpose of the study is to determine whether pharmacist provision of a bridging supply of a progestogen only pill (POP) plus rapid access to a local SRH clinic, results in increased uptake of effective contraception and prevents more unintended pregnancies than provision of EC alone. This is a well-planned study which holds promise for answering important questions about optimizing care for women who seek emergency contraception at pharmacies. Comments on the manuscript:

1) The status of the study is provided at the end; i.e. the date enrollment started and anticipated study end; would be helpful to have this earlier in the manuscript (i.e. abstract)

Response: We have provided the date in the section as advised by the BMJ open reporting requirements.

2) It would be helpful if the introduction, the methods, or discussion included a section which describes consideration given to why cluster randomization is necessary and the risk of the intervention effect from one period carrying over to the control period.

Response: The cluster design was felt necessary for logistical reasons. Many of the pharmacies included are single-handed, and run as businesses, in which responding to customers quickly is key. It is felt – and confirmed in the qualitative work in the pilot study - that an individually randomised trial would simply not recruit, as it was not feasible to take 2-3 minutes out to randomise each individual. The crossover nature of the cluster design was chosen for efficiency, and by having a different set of women recruited at a pharmacy in the two different periods we avoided contamination by the participant. The purpose of the washout out period (was to minimise intervention effect carrying over from one period to another, as part of any contamination effect mediated by the pharmacist. We have added this to discussion section.

‘The cluster design was felt necessary for logistical reasons and confirmed in the qualitative work of our pilot study [17, 23] that an individually randomised trial would simply not recruit, as it was not feasible for pharmacists within a busy pharmacy to take additional time to randomise each individual. The crossover nature of the cluster design was chosen for efficiency, and by having a different set of women recruited at a pharmacy in the two different periods we avoided contamination by the participant. The purpose of the washout out period (was to minimise intervention effect carrying over from one period to another, as part of any contamination effect mediated by the pharmacist. With the cluster crossover design, each cluster will act as its own control and fewer pharmacies are required than with a parallel cluster design.’

3) An explanation of how the methods for obtaining the WPC and the BPC were chosen for the sample size calculation would be helpful for readers (i.e. why use 4-month data of 500 subjects?), and potential factors which might influence the magnitude of the variation for the estimates .

Response: This was a key challenge in estimating the sample size. There was a paucity of data on cluster crossover designs – either in emergency contraception settings or more generally community pharmacy -based studies that we could draw on. So at the design stage we assumed a range of correlations (WPC and BPC) what were felt to be plausible, with the intention of re-estimating the sample size when we had data within the study to calibrate the assumed WPC and BPC. This was set at when we had mature data at 4 months on the first 500 randomised. In practice, the study was re-

designed and the co-primary outcomes of 4-month use of contraception and 12-month abortion was simplified to just a single primary outcome of 4-month use of contraception, due to severe problems in recruitment, and the need by the funder to complete the study in a reasonable time frame without further extensive costs to the public purse, so in this eventuality the re-estimation of the sample size was not undertaken. This issue of the challenge of informing the assumptions for the sample size of a cluster crossover design with relevant external data is a key learning point from the study, which will be fully discussed in reporting the findings.

Reviewer: 4

The authors have done an admirable job of designing and clearly explaining the statistical aspects of the proposed Bridge-it trial. I do have some minor concerns, questions and suggestions:

1. WPC and BPC are described as components of variability in the paragraph labeled participants, and then, more accurately (in my view), as correlations in the paragraph labeled sample size calculation. I suggest making these descriptions consistent.

Response: We agree and have consistently now described them as correlations.

2. It might be worthwhile to specify the range of values of these two correlations being considered. For what it's worth, my estimates of the power of the study for the proposed range of sample sizes were ~90% for values of WPC <5%, and BPC no more than 2 points less than WPC.

Response: We thank the reviewer for this suggestion and have followed their recommendation for calculated power of 90% for the ranges of WPC and BPC as suggested.

3. The updated sample size calculation will correctly use the hypothesized treatment effect, not an internal estimate. While it makes good sense using study data to update preliminary estimates of WPC and BPC after 4-month outcomes for 500 participants have been obtained, omitting treatment from the model used to estimate the variance components could inflate both correlations at least slightly, although this bias might be negligible compared to sampling error. Will the WPC and BPC estimates be obtained without unblinding, to avoid inflating the type-I error rate (or at any rate giving the appearance of having done so)?

Response: We agree with the reviewer that we must use the hypothesised treatment effect, and not be updated by an internal estimate based on an emerging treatment effect for the data already matured at the time of the sample size re-estimate. For the updated estimates of WPC and BPC, it is however essential to use the updated data. We agree that ideally that an indicator variable for treatment would improve the model estimating the variance components, but preferred to have the simplicity and security of not having to unblind any analysis at this stage, hoping as pointed out by the reviewer that the magnitude of any reduction in variability by including the treatment effect would be small in the context of the overall level of variability. We will discuss this issue fully in a technical appendix when reporting the findings.

4. The Bayesian approach of Turner et al. is developed in the context of multivariate outcomes and thus would apply to the secondary process outcomes that are to be collected. However, it is less clearly suited to the univariate primary outcome of contraceptive use at 4 months. One concern is that Bayesian methods may not ultimately be acceptable to reviewers expecting frequentist methods for an RCT, in part because, as Turner et al. point out, the results can be sensitive to the choice of priors. If Bayesian analysis of the primary outcome is intended, I think this should be made explicit.

Response: We will include details of this issue in the Statistical Analysis Plan. The reviewer makes an interesting point about the original development of Turner's analysis being for multivariate outcomes. We are currently looking into this, before we see any unblinded data, and if we find that it isn't as suitable as we thought for the univariate primary outcome of contraceptive use at 4 months, we will modify our analysis plans. We agree that there is often difficulty from Reviewers fairly assessing

Bayesian analyses, but in the particular issue of choice of priors, the lack of quality evidence would suggest that non-informative priors are an appropriate choice here, which usually reduces the problem. We would potentially include different priors to demonstrate robustness of our findings to the assumptions in suitably specified sensitivity type analyses.

5. In addition, the analysis plans should probably mention some standard checks for crossover trials such as checks for order effects and order-treatment interaction (recognizing that power for the latter is low).

Response: Thank you for this suggestion. We will incorporate these standard checks in our analysis plan. Full details will be included in the Statistical Analysis Plan.

6. I didn't understand the proposal at the end of the paragraph on quantitative analyses for a subgroup analysis of effect modification by LARC vs non-LARC, which seem like two possible forms of the binary outcome of contraceptive use at 4 months, not baseline characteristics (as usual for effect modifiers). If so, wouldn't it make sense to analyze this as a multinomial outcome, comparing the treatment effects on LARC vs no use and non-LARC vs no use?

Response: Good point, thank you for spotting this. We agree that subgroups should be defined using baseline data, and as such this doesn't qualify. Indeed, we are looking at a 3-level outcome – None, LARC, and non-LARC. We will as suggested use an appropriate multinomial outcome and construct the 2 comparisons as suggested.

'Subgroup analyses will explore the possible effect modification by LARC (most effective contraceptive methods) vs non-LARC vs no use of contraception'

7. The plans for sensitivity analyses examining robustness to missing data, in particular non-ignorable missingness, are excellent, and will be important if the expected 25% of participants do not provide the primary outcome at 4 months. Still, that is a rather large loss to follow-up rate. Could something be done to reduce it, such as provision of stipends at 4 months as well as baseline?

Response : The attrition rate in the pilot study was high (61% follow up at 8 wks, Michie et al 2014) , on which we based this study. We fully aim for a lower loss to follow up rate. In order to achieve this we will adopt a pan study approach to minimising attrition, informed by combined experience of the team and evidence in the literature including cochrane reviews on strategies to minimise attrition. This includes excellent training of study staff (pharmacists, nurses, researcher), good participant screening, excellent information for women about study requirements, no question unanswered, good communication keeping study requirements for women as simple as possible and use of incentives. We chose to provide the incentive voucher at recruitment as funding was limited. The study research nurses will have excellent communication skills and a flexible approach to follow up inc. evening follow up phone calls/ texting according works best for participants.

We have developed a simple follow up for the study using text messaging/ phone, to keep it as easy as possible for women to continue to participate in the study. Finally, we will use state of the art analytics to provide the best evidence from this study.

VERSION 2 – REVIEW

REVIEWER	Rebecca Simmons, PhD University of Utah, United States of America
REVIEW RETURNED	28-Aug-2019
GENERAL COMMENTS	No remaining comments. Authors still have not fully discussed limitations to this study in the context of the protocol, but perhaps that is a limitation of the type of article and this will be covered

	elsewhere.
REVIEWER	Tina Raine-Bennett MD, MPH Kaiser Permanente Northern California United States
REVIEW RETURNED	19-Aug-2019
GENERAL COMMENTS	The authors have adequately addressed reviewers comments; the revised draft is clearer and the protocol holds promise to answer important research questions.
REVIEWER	Eric Vittinghoff University of California, San Francisco, California, USA
REVIEW RETURNED	26-Aug-2019
GENERAL COMMENTS	I thank the authors for being responsive to the concerns I raised. My only remaining quibble is with the proposed subgroup analysis exploring effect modification by LARC -- again, I don't see how you can have modification of the treatment effect by an outcome -- it has to be by a baseline covariate.

VERSION 2 – AUTHOR RESPONSE

Reviewer: 4

Reviewer Name: Eric Vittinghoff

Institution and Country: University of California, San Francisco, California, USA

Please state any competing interests or state 'None declared': None declared

Please leave your comments for the authors below

I thank the authors for being responsive to the concerns I raised. My only remaining quibble is with the proposed subgroup analysis exploring effect modification by LARC -- again, I don't see how you can have modification of the treatment effect by an outcome -- it has to be by a baseline covariate.

Response : We previously agreed with what the reviewer was suggesting and changed – see reviewers suggestion highlighted in yellow

Same reviewer previously said : “I didn't understand the proposal at the end of the paragraph on quantitative analyses for a subgroup analysis of effect modification by LARC vs non-LARC, which seem like two possible forms of the binary outcome of contraceptive use at 4 months, not baseline characteristics (as usual for effect modifiers). If so, wouldn't it make sense to analyze this as a multinomial outcome, comparing the treatment effects on LARC vs no use and non-LARC vs no use?”

Response: Good point, thank you for spotting this. We agree that subgroups should be defined using baseline data, and as such this doesn't qualify. Indeed, we are looking at a 3-level outcome – None, LARC, and non-LARC. We will as suggested use an appropriate multinomial outcome and construct the 2 comparisons as suggested.

We have amended text to : ‘Subgroup analyses will explore the possible effect modification by LARC (most effective contraceptive methods) vs non-LARC vs no use of contraception’

Reviewer: 1

Reviewer Name: Rebecca Simmons, PhD

Institution and Country: University of Utah, United States of America

Please state any competing interests or state ‘None declared’: None declared

Please leave your comments for the authors below

No remaining comments. Authors still have not fully discussed limitations to this study in the context of the protocol, but perhaps that is a limitation of the type of article and this will be covered elsewhere.

Response: Correct- this is a protocol and is more appropriate that the study with results will discuss limitations in full.

We hope that the responses will be satisfactory and that this protocol may now be acceptable for publication in BMJ open